# Effect of Tool Design and Process Parameters on Lap Joints Made by Right Angle Friction Stir Welding (RAFSW)

**Mahboubeh Momeni and Michel Guillot *** 

PI2/REGAL Research Team, Department of Mechanical Engineering, Laval University,
Québec, QC G1V 0A6, Canada
* Correspondence: mguillot@gmc.ulaval.ca; Tel.: +1-(418)-998-6549 or +1-(418)-656-3343

**Abstract:** In recent decades, friction stir welding (FSW) has attracted extensive attention of academic and industrial sectors as the most considerable development in metal joining processes. FSW lap joint is an interesting alternative for rivets, fusion welds and bonding particularly in the transportation industry. In this paper, the effect of tool design and process parameters on the generated downward axial force and strength of AA6061-T6 lap joints is studied. The welds are made by a low-cost friction stir welding technique at right angle (RAFSW). The studied tool design parameters are shoulder diameter, shoulder groove depth, pin length, pin angle, pin base diameter and pin lead. Moreover, the effect of tool rotational speed, traverse speed, plunge depth and lap joint configuration is evaluated. The Taguchi method is used to design the experiments and artificial neural network (ANN) modeling is applied to predict the plunging force and the strength of the joints. The results indicate that a quality weld can be obtained at low downward axial forces during welding by proper selection of tool design and process parameters. It is identified that one can achieve a quality lap joint at traverse speeds as high as 1400 mm/min and downward axial forces as low as 3.2 kN by a low-cost RAFSW technique.

**Keywords:** friction stir welding; lap joint; tool design; process parameters; Taguchi design of experiments; artificial neural network modeling

## 1. Introduction

Friction stir welding (FSW) is a newly emerged solid-state joining process. In the past decades, it has increasingly attracted the interest of researchers and industry owing to its prominent advantages over fusion welding techniques. It is a green, versatile technology capable of making high quality welds. However, FSW equipment costs and royalties limit the use of this technique in current industry [1–4]. Some attempts are made to overcome this issue. In this regard, a low-cost friction stir welding technique at right angle (RAFSW) has been developed by our research team, recently [5,6]. It can be employed on low-cost 3-axis CNC machining centers without any need of prior modification of the machine. The RAFSW technique is capable of making sound, defect-free welds at a zero tilt angle with low axial forces during welding compared to common FSW techniques. Thus, this method not only works with 3-axis machines instead of 4- or 5-axis machines, but also does not need sturdy, stiff and high capacity expensive equipment. Moreover, clamping and fixturing can easily use existing vises and clamps of the CNC machining center since the required holding force is lower [5,6]. Until now, FSW has primarily been employed to produce butt joints [7,8] and sometimes for a lap joint. By developing FSW techniques for lap joints, the number of applications of this technique would expand, drastically. Assembly of parts and components in the transportation industry is widely done using lap joint

configuration [8–10]. This configuration is extensively used in mechanical structures in format of riveted joints and fusion welded joints. Taking advantage of FSW lap joints, instead of the mentioned joints, can lead to reduction of weight, cost and production time of the joints. Moreover, it boosts the mechanical properties of the joint since the joint has less defects and imperfections than other types of joining processes [7,11,12].

Production of sound, quality lap joints is not with the same ease of making butt joints due to some reasons [13,14]. Firstly, in overlap joints, there are two crack-like unwelded zones that can act as crack initiation sites when the joint is under load [8,11,13]. Moreover, there are two types of defects in lap joints, which are hooking and cold lap defects. Hook defect is not always the fracture initiation site and sometimes is not really bad especially for dissimilar lap joints [15–17] but sometimes the fracture starts from theses defects [9,13]. In general, they can have damaging effect on the strength of the welds. Their adverse effect can be restricted and even avoided by proper tool design and process parameters [11,13,14]. Furthermore, the disruption of the oxides at the sheets interfaces is more difficult in the lap joint configuration [13]. In addition, the weld width plays a considerable role in the joint performance [14]. The wider the width of the weld, the more the downward axial forces generated during the welding process, which is not desirable as it increases the cost of equipment and fixturing. Moreover, plate thinning and entrapment of oxide particles happen in the lap joints, which must be minimized [8,11,13]. Additionally, in lap joints, there are two overlapped sheets, which tend to separate from each other when the FSW tool progresses in the material. That can be avoided by more clamping and fixturing of the structure or by applying some changes in the tool design compared to the tools for butt joints. The negative effects of the aforementioned problems can be prevented by employing a proper tool design and adequate process parameters [10,11,13]. Thus, the presence of these defects becomes negligible.

Although, there are numerous studies on the effect of tool design and process parameters [7,9] on the quality of the butt joints made by FSW technique, there is less research on the lap joints [8]. Some research has shown the effect of tool design including the shoulder shape (convex, flat or concave), shoulder dimension, pin shape (cylindrical or conical) and surface features on the shoulder and pin (like flutes, grooves or threads) on the quality of the lap joints [11,13,18]. For instance, Yue et al. have identified that a reverse-threaded pin works better than a threaded pin to make quality AA2024 lap joints [19]. Buffa et al. reported that the effective material flow greatly depends on the tool design. Cylindrical-conical pins were the most effective design to make 2198-T4 lap joints in their research [10]. Some other research has investigated the impact of the process parameters including the tool plunge depth, rotational speed, traverse speed and configuration of the lap joint on the quality of the welds [11,12,20,21]. For example, the effect of the process parameters on the mechanical properties of AA5456 lap joints has been studied in a research. The results show that the optimal mechanical properties are obtained when the rotational and traverse speed are 250 rpm and 75 mm/min, respectively [7]. In another study investigating the effect of process parameters on AA6060 lap joints, it was shown that the increase of the rotation speed causes the decrease of joint strength [22]. In the majority of these studies, the traverse speed is too low for industrial purposes [7,8,21,22]. Additionally, it was demonstrated that the lap joint configuration affects the strength of the joints [10,11] and that the best results were found when the advancing side of the weld is located on the upper sheet of the lap joint [10].

Nowadays, there is an increasing interest to employ FSW techniques for lap joint assembly in the transportation industry. AA6061-T6 like some other alloys has many examples of application such as ship hulls, truck roof and side panels, wagon roofs. Unfortunately, there is a lack of information and too few studies on the determination of the effective working window of the tool design parameters and the process parameters to make quality AA6061-T6 lap joints by the FSW technique. To make an extensive study on the effect of these parameters on the quality of joints, Taguchi method can be employed to minimize the number of experiments [23]. Besides, artificial neural network modeling can be used as a powerful tool to predict the behavior of the joints based on the experimental data [24,25].

These methods have successfully been applied in many researches regarding the FSW joints mainly for butt joint configuration [5,8,25,26].

In this paper, an extensive study is conducted on the effect of tool design parameters and process parameters on the quality of AA6061-T6 lap joints made by RAFSW technique. To this end, the Taguchi method is used to design the experiments. Afterwards, artificial neural network (ANN) modeling is employed to predict the effect of the mentioned parameters on the downward axial force during the welding process and the strength of the joints represented in terms of fracture force. In this research, high traverse speeds are applied to make the process promising for industrial use.

## 2. Materials and Methods

In this paper, the RAFSW technique was employed to make the lap joints of AA6061-T6 extruded sheets of 1.6 mm thickness. Table 1 presents the chemical composition and tensile strength of this sheet metal according to the specification provided by the extruder. The sheets were cut to pieces with the dimensions of 245 mm × 88 mm × 1.6 mm using a shear press. A 3-axis CNC machining center, Fryer MC-15, with 25 HP spindle using CAT40 tool holders was used in this research for friction stir welding. The maximum rotational speed and the axis peak trust of this machine were 8000 rpm and 15 kN, respectively. To make the joints, two sheets were fixed on top of each other on a rigid back-plate installed on top of a calibrated dynamometer. The dynamometer was a 3-axis Kistler 9265B. The RAFSW tool was mounted into a long reach tool holder. In this paper, the specially designed tools to make RAFSW lap joints were flat shoulder tools with some grooves on the shoulder. The pin was threaded and had a conical shape. Single-pass welds were conducted along the extruded direction of the sheets. Figure 1 depicts a general view of the tool to make lap joints by RAFSW, the RAFSW set-up, clamping of the sheets on the Kistler dynamometer, the tool in the tool holder and a welded sample.

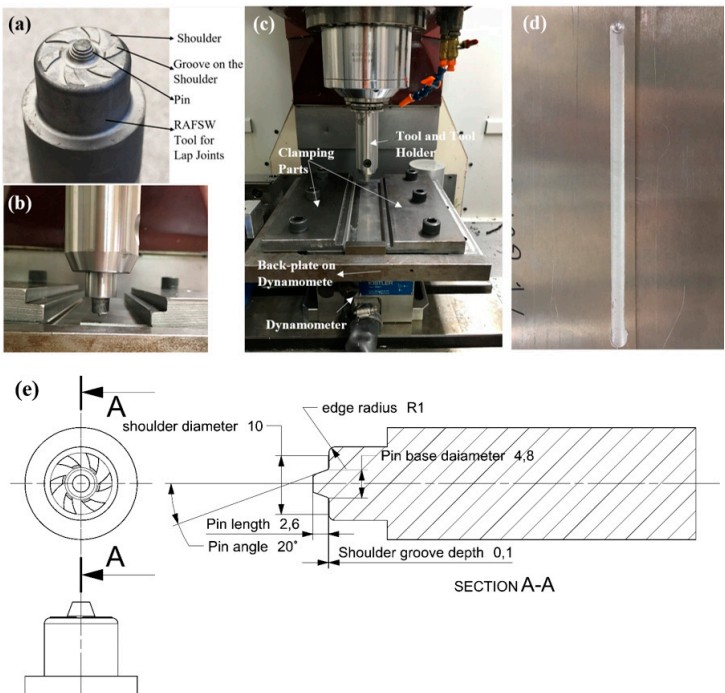

**Figure 1.** (**a**) The tool shape to make lap joints by the friction stir welding technique at right angle (RAFSW) technique; (**b**) The close view of the tool and tool holder; (**c**) The RAFSW set-up including the tool, tool holder, clamped sheets on the back-plate and the back-plate installed on the dynamometer; (**d**) The welded sample; (**e**) The schematic of the tool and the design parameters (the pin lead, not illustrated, is the distance between the threads on the pin).

To make the lap joints, there are two types of configurations, which are not of the same properties due to the asymmetric nature of FSW joints. Thus, both types of configurations were studied in this research as illustrated in Figure 2. The fracture force of the joints was investigated by the tensile shear test. The weld coupons were machined with the dimensions specified in Figure 3. The top sheet was loaded in the tensile shear test [12]. A hydraulic testing machine was used to conduct the uniaxial tensile tests under a crosshead speed of 1 mm/min. The machine was equipped with a load cell of 44.5 kN calibrated to ±0.08% In this research, the single lap shear tests were done without spacers because the goal of this paper was to establish the strength of the lap joints for the applications such as the assembly of truck panels, bus and wagon roofs. In such applications, the force applied to the joint was not centered. Therefore, the single lap shear test without spacers could replicate the applied forces closer to reality than the test with spacers.

**Table 1.** Chemical composition and tensile strength of the base metal.

| | Chemical Composition (wt%) | | | | | | |
|---|---|---|---|---|---|---|---|
| Material | Al | Mg | Mn | Cu | Fe | Si | *UTS* (MPa) |
| AA-6061-T6 | Bal. | 0.83 | 0.07 | 0.19 | 0.19 | 0.55 | 285 |

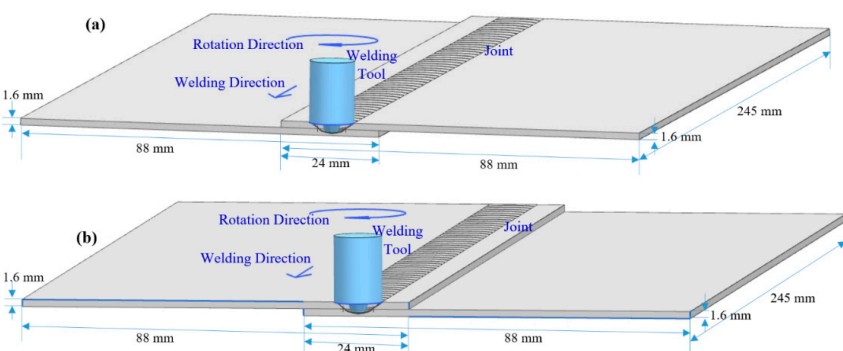

**Figure 2.** The schematic of the welding process for two possible configurations of the lap joints; (**a**) the top sheet on the right side (configuration 1); (**b**) the top sheet on the left side (configuration 2).

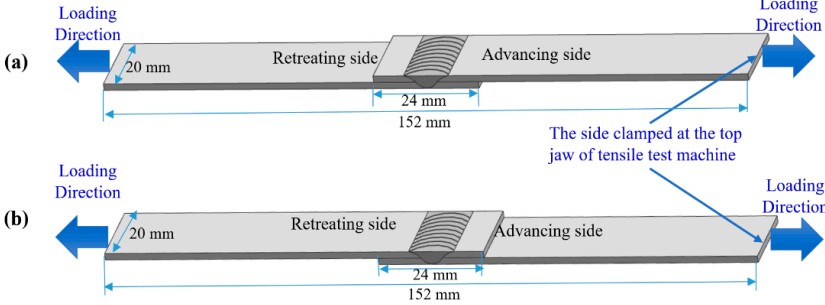

**Figure 3.** The schematic and dimensions of the weld coupons to make the tensile test. (**a**) Configuration No. 1, when the advancing side is on the upper sheet; (**b**) Configuration No. 2, when the retreating side is on the upper sheet.

## 3. Results and Discussion

### 3.1. Design of Experiments

The design of experiments (DOE) was performed based on the Taguchi method. Accordingly, a L16 orthogonal array was used to explore the effect of the tool geometry and the process parameters on the downward axial force and the fracture force of the lap joints made by the RAFSW technique. The tool geometry parameters were the shoulder diameter, the shoulder groove depth, the pin length, the pin angle, the pin base diameter and the pin lead as shown in Figure 1. Like industrial tools,

a 1 mm radius was added to the edge of all tool shoulder to minimize burr formation and improve welding when imperfect materials and assemblies were present. The process parameters were the tool traverse and rotational speeds, the tool plunge depth and the lap joint configuration. Therefore, 16 experiments were designed based on the L16 array. In addition, for each test of the array two passes named "a" and "b" were done. In the first pass, the plunge depth was set exactly to the pin length. In the second pass, the tool plunge depth was deeper by 0.05 mm and 0.08 mm for tool groove depths of 0.1 mm and 0.25 mm, respectively. The designed L16 arrays and both sets experimental results are presented in Table 2. In addition, some additional experiments were conducted to provide more data for the modeling purposes, as shown in Table 3. The tools made for all experiments are illustrated in Figure 4. Overall, 20 tools were made and 40 experiments were conducted in the experimental section of this research.

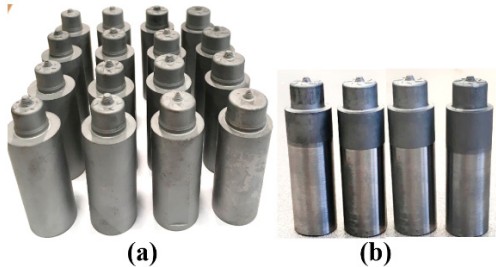

**(a)**        **(b)**

**Figure 4.** (**a**) 16 tools made according to the L16 design of experiments (DOE) presented in Table 2; (**b**) Four tools made regarding Table 3.

**Table 2.** The experiments conducted according to the L16 orthogonal array. For each condition of the L16 array, tow tests were done with different tool plunge depths.

| Sample No. | Sample Code | PL (mm) | SD (mm) | SGD (mm) | PBD (mm) | PA (°) | PLD (mm) | V (mm/min) | w (rpm) | PD (mm) | C | DAF (N) | FF (N) |
|---|---|---|---|---|---|---|---|---|---|---|---|---|---|
| 1 | 1-a | 1.8 | 8.5 | 0.1 | 4 | 20 | 0.25 | 1400 | 3500 | 1.8 | 1 | 2600 | 2994 |
| 2 | 1-b | 1.8 | 8.5 | 0.1 | 4 | 20 | 0.25 | 1400 | 3500 | 1.85 | 1 | 3060 | 4453 |
| 3 | 2-a | 1.8 | 9.5 | 0.1 | 4 | 20 | 0.45 | 2000 | 5000 | 1.8 | 2 | 2500 | 3390 |
| 4 | 2-b | 1.8 | 9.5 | 0.1 | 4 | 20 | 0.45 | 2000 | 5000 | 1.85 | 2 | 3120 | 3937 |
| 5 | 3-a | 1.8 | 10.5 | 0.25 | 4.8 | 24 | 0.25 | 1400 | 3500 | 1.8 | 2 | 2900 | 5373 |
| 6 | 3-b | 1.8 | 10.5 | 0.25 | 4.8 | 24 | 0.25 | 1400 | 3500 | 1.88 | 2 | 4130 | 5422 |
| 7 | 4-a | 1.8 | 11.5 | 0.25 | 4.8 | 24 | 0.45 | 2000 | 5000 | 1.8 | 1 | 2810 | 2847 |
| 8 | 4-b | 1.8 | 11.5 | 0.25 | 4.8 | 24 | 0.45 | 2000 | 5000 | 1.88 | 1 | 4320 | 5218 |
| 9 | 5-a | 2.2 | 8.5 | 0.1 | 4.8 | 24 | 0.25 | 2000 | 5000 | 2.2 | 2 | 3210 | 3510 |
| 10 | 5-b | 2.2 | 8.5 | 0.1 | 4.8 | 24 | 0.25 | 2000 | 5000 | 2.25 | 2 | 3590 | 3390 |
| 11 | 6-a | 2.2 | 9.5 | 0.1 | 4.8 | 24 | 0.45 | 1400 | 3500 | 2.2 | 1 | 3420 | 5218 |
| 12 | 6-b | 2.2 | 9.5 | 0.1 | 4.8 | 24 | 0.45 | 1400 | 3500 | 2.25 | 1 | 4000 | 5609 |
| 13 | 7-a | 2.2 | 10.5 | 0.25 | 4 | 20 | 0.25 | 2000 | 5000 | 2.2 | 1 | 3200 | 4502 |
| 14 | 7-b | 2.2 | 10.5 | 0.25 | 4 | 20 | 0.25 | 2000 | 5000 | 2.28 | 1 | 3840 | 5756 |
| 15 | 8-a | 2.2 | 11.5 | 0.25 | 4 | 20 | 0.45 | 1400 | 3500 | 2.2 | 2 | 3310 | 2709 |
| 16 | 8-b | 2.2 | 11.5 | 0.25 | 4 | 20 | 0.45 | 1400 | 3500 | 2.28 | 2 | 4530 | 3292 |
| 17 | 9-a | 2.6 | 8.5 | 0.25 | 4 | 24 | 0.45 | 1400 | 5000 | 2.6 | 2 | 2970 | 4083 |
| 18 | 9-b | 2.6 | 8.5 | 0.25 | 4 | 24 | 0.45 | 1400 | 5000 | 2.68 | 2 | 3240 | 3314 |
| 19 | 10-a | 2.6 | 9.5 | 0.25 | 4 | 24 | 0.25 | 2000 | 3500 | 2.6 | 1 | 4100 | 3358 |
| 20 | 10-b | 2.6 | 9.5 | 0.25 | 4 | 24 | 0.25 | 2000 | 3500 | 2.68 | 1 | 4720 | 4746 |
| 21 | 11-a | 2.6 | 10.5 | 0.1 | 4.8 | 20 | 0.45 | 1400 | 5000 | 2.6 | 1 | 3170 | 6107 |
| 22 | 11-b | 2.6 | 10.5 | 0.1 | 4.8 | 20 | 0.45 | 1400 | 5000 | 2.65 | 1 | 3780 | 5640 |
| 23 | 12-a | 2.6 | 11.5 | 0.1 | 4.8 | 20 | 0.25 | 2000 | 3500 | 2.6 | 2 | 4480 | 2171 |
| 24 | 12-b | 2.6 | 11.5 | 0.1 | 4.8 | 20 | 0.25 | 2000 | 3500 | 2.65 | 2 | 4700 | 2300 |
| 25 | 13-a | 3 | 8.5 | 0.25 | 4.8 | 20 | 0.45 | 2000 | 3500 | 3 | 1 | 4320 | 4982 |
| 26 | 13-b | 3 | 8.5 | 0.25 | 4.8 | 20 | 0.45 | 2000 | 3500 | 3.08 | 1 | 4680 | 6557 |
| 27 | 14-a | 3 | 9.5 | 0.25 | 4.8 | 20 | 0.25 | 1400 | 5000 | 3 | 2 | 2930 | 2123 |
| 28 | 14-b | 3 | 9.5 | 0.25 | 4.8 | 20 | 0.25 | 1400 | 5000 | 3.08 | 2 | 3360 | 2042 |
| 29 | 15-a | 3 | 10.5 | 0.1 | 4 | 24 | 0.45 | 2000 | 3500 | 3 | 2 | 3934 | 2136 |
| 30 | 15-b | 3 | 10.5 | 0.1 | 4 | 24 | 0.45 | 2000 | 3500 | 3.05 | 2 | 4390 | 2056 |
| 31 | 16-a | 3 | 11.5 | 0.1 | 4 | 24 | 0.25 | 1400 | 5000 | 3 | 1 | 3040 | 3581 |
| 32 | 16-b | 3 | 11.5 | 0.1 | 4 | 24 | 0.25 | 1400 | 5000 | 3.05 | 1 | 3750 | 4141 |

Abbreviation description: Pin length (*PL*), Shoulder Diameter (*SD*), Shoulder Groove Depth (*SGD*), Pin base diameter (*PBD*), Pin angle (*PA*), Pin lead (*PLD*), Traverse speed (*V*), Rotation speed (*w*), Plunge Depth (*PD*), Configuration (*C*), Downward axial force (*DAF*), Failure force (*FF*).

**Table 3.** Some more experiments conducted to explore more regarding the effect of tool and process parameters on downward axial force and fracture force.

| Sample No. | Sample Code | PL (mm) | SD (mm) | SGD (mm) | PBD (mm) | PA (°) | PLD (mm) | V (mm/min) | w (rpm) | PD (mm) | C | DAF (kN) | FF (kN) |
|---|---|---|---|---|---|---|---|---|---|---|---|---|---|
| 33 | 17-a | 1.8 | 8.5 | 0.1 | 3.5 | 20 | 0.25 | 1700 | 4250 | 1.8 | 1 | 2970 | 3118 |
| 34 | 17-b | 1.8 | 8.5 | 0.1 | 3.5 | 20 | 0.25 | 1700 | 4250 | 1.835 | 1 | 3320 | 3982 |
| 35 | 18-a | 1.75 | 12 | 0.3 | 5 | 25 | 0.5 | 1800 | 4500 | 1.75 | 2 | 2760 | 5454 |
| 36 | 18-b | 1.75 | 12 | 0.3 | 5 | 25 | 0.5 | 1800 | 4500 | 1.83 | 2 | 4320 | 5961 |
| 37 | 19-a | 1.75 | 12 | 0.3 | 4.5 | 25 | 0.35 | 2200 | 5000 | 1.75 | 1 | 3340 | 2224 |
| 38 | 19-b | 1.75 | 12 | 0.3 | 4.5 | 25 | 0.35 | 2200 | 5000 | 1.83 | 1 | 4600 | 4043 |
| 39 | 20-a | 1.75 | 11.5 | 0.3 | 4.5 | 25 | 0.45 | 2400 | 5500 | 1.75 | 2 | 3020 | 4328 |
| 40 | 20-b | 1.75 | 11.5 | 0.3 | 4.5 | 25 | 0.45 | 2400 | 5500 | 1.83 | 2 | 4000 | 4582 |

### 3.2. Artificial Neural Network Modeling

The experimental data presented in the previous section were utilized to train artificial neural networks in this section. Feed-forward neural networks with backpropagation algorithm were used to model the relationship between the tool geometry and the process parameters on the downward axial force and the fracture force of the joints. Indeed, several factors affect the prediction accuracy of the backpropagation neural network models such as architecture of the network, momentum coefficient and learning rate of the model [24,27]. Accordingly, in this paper, the effect of these parameters on the accuracy of the neural network models was studied to find the best ANN modeling factors. An ANN model with too small architecture can result in an insufficient degree of freedom; and too large of a network causes it to over fit the data. Thus, there are an optimized number of neurons and hidden layers to have a reliable ANN model [28]. In this research, several numbers of neurons in one hidden layer were evaluated to find the optimal architecture. The studied factors to compare the different architectures were root-mean squared error (*RMSE*), maximum error, mean relative error (*MRE*) and mean absolute error (*MAE*) [25]. Moreover, the effect of the learning rate and the momentum coefficient in the accuracy of the models was studied. The correlation of the learning rate, the momentum coefficient with *RMSE* and maximum error are illustrated in Figure 5. To keep the *RMSE* and maximum error minimized, the optimal learning rate and momentum coefficient was 0.5 for both of them. Additionally, it was found that the optimal architecture was 10-8-1 for both models of downward axial forces and fracture forces shown in Figure 6. Table 4 presents the employed condition to make the final ANN models in this paper.

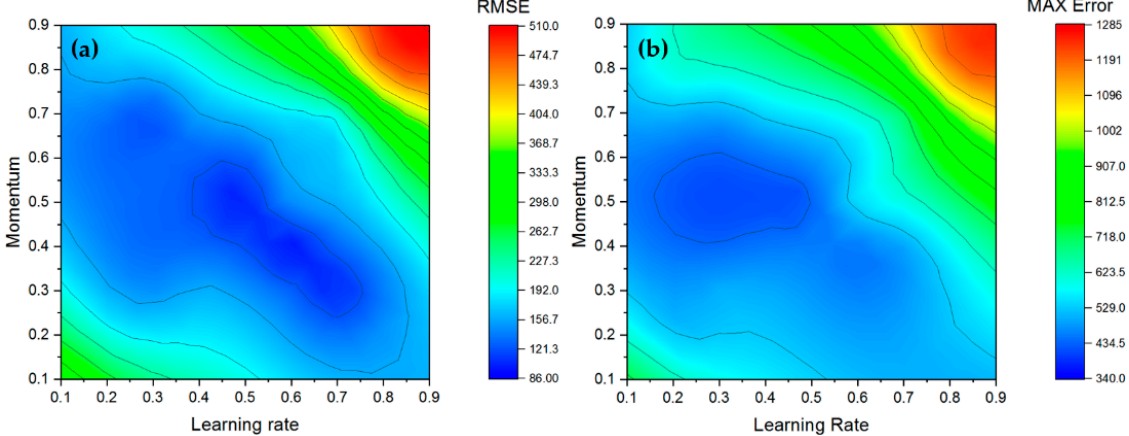

**Figure 5.** The effect of the learning rate and momentum coefficient on (**a**) Root-mean squared error (RMSE); (**b**) Maximum error when the architecture of the model is 10-8-1.

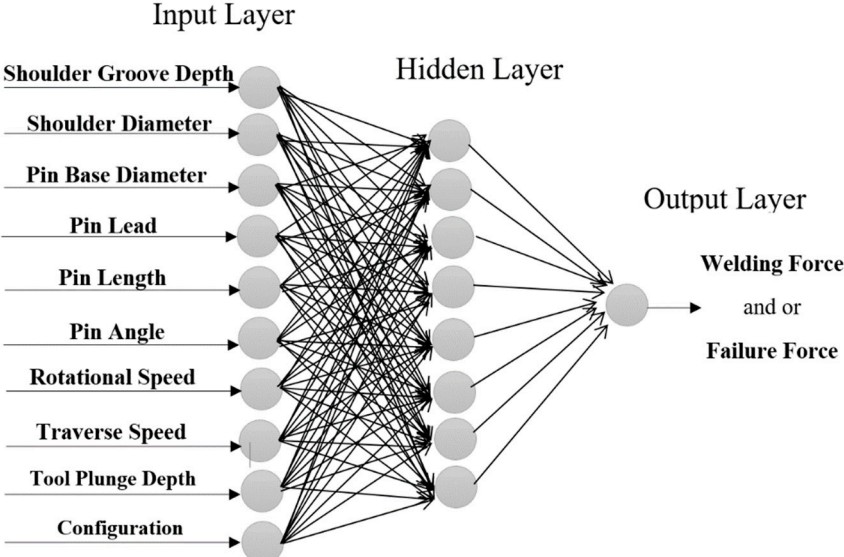

**Figure 6.** The architecture of artificial neural network (ANN) models in this paper. The architecture of the model for the downward axial force and fracture force was 10-8-1 for both of them.

**Table 4.** The details regarding the developed ANN models in this paper to study the effect of tool geometry and process parameters on the downward axial force and failure force.

|  | **Welding Force Model** | **Failure Force Model** |
|---|---|---|
| Network configuration | 10-8-1 | 10-8-1 |
| Number of inputs | 10 | 10 |
| Number of hidden layers | 1 | 1 |
| Number of neurons | 8 | 8 |
| Number of outputs | 1 | 1 |
| Total no. of experimental data | 40 | 40 |
| Learning rate | 0.5 | |
| Momentum | 0.5 | |
| Method | Back propagation algorithm | |

The experimental data of the 40 tests obtained in the previous section were used to develop the ANN models, in this section. Thirty-six experiments were separated from 40 experiments as training data for the ANN models; and the four remaining tests were utilized as confirmation tests to evaluate the predictability and the accuracy of the developed ANN models. Table 5 shows the *RMSE* and maximum error of the developed 10-8-1 ANN models for training data, confirmation data and the entire set of the data. It is important to keep the error of the both trained data and confirmation data minimized. The *RMSE*, *MRE*, *MAE* and maximum error of the entire set of experiments for the developed ANN models are presented in Table 6. From Tables 5 and 6, it is concluded that the amount of different kinds of errors for the developed models was low enough to make them capable of accurate predictability. Furthermore, the relationship between the experimental data and predicted values by the developed ANN models was studied by calculating the amount of correlation coefficient ($R^2$) using linear regression analysis. When the correlation coefficient is close to one, it indicates a close relationship between the experimental data and predicted data by the developed ANN models [24,25,29]. According to Table 7, the correlation coefficient for training data, confirmation data and the overall data were respectively 0.999981092, 0.998622385 and 0.999846253 for the ANN model of downward axial force. For the failure force model, they were 0.999722461, 0.998734858 and 0.999634318, respectively. These coefficient values confirmed the accuracy of both downward axial force and failure force models. The comparison between experimental and predicted values, shown in Table 8, indicated errors of less than 5% in all cases. In summary, Tables 5–8 validated the reliability of

the developed ANN models to show the effect of the tool geometry and the process parameters of the downward axial force and the fracture force of the lap joints made by the RAFSW technique.

**Table 5.** The *RMSE* and maximum error for the training data, the confirmation data and the overall data when the architecture of the neural networks is 10-8-1.

| Training Data | | | | Confirmation Data | | | | All Data | | | |
|---|---|---|---|---|---|---|---|---|---|---|---|
| Downward axial force | | Failure force | | Downward axial force | | Failure force | | Downward axial force | | Failure force | |
| *RMSE* | *Max E.* | *RMSE* | *Max E.* | *RMSE* | *Max E.* | *RMSE* | *Max E.* | *RMSE* | *Max E.* | *RMSE* | *Max E.* |
| 15.97 | 52 | 71.49 | 211 | 135.76 | 169 | 129.53 | 174 | 45.53 | 169 | 80.96 | 211 |

**Table 6.** The amount of different kind of errors for the developed ANN models.

| | RSME | MAE | MRE | Maximum Error |
|---|---|---|---|---|
| Formula | $\left(\frac{1}{N}\sum_{i}^{N}(A_i - Y_i)^2\right)^{1/2}$ | $\frac{1}{N}\sum_{i}^{N}\lvert A_i - Y_i\rvert$ | $\frac{1}{N}\sum_{i}^{N}\left(\frac{\lvert A_i - Y_i\rvert}{A_i}\right)\times 100$ | $\lvert A_i - Y_i\rvert$ |
| Downward axial force model | 45.53 | 21.61 | 0.64% | 169 |
| Failure force model | 80.96 | 61.85 | 1.73% | 211 |

$A_i$, $Y_i$, $N$ and $i$ are the experimental value, predicted value, total number of experimental data and trial number, respectively. The formulas are from [25].

**Table 7.** The amount of correlation coefficient ($R^2$) for the training data, confirmation data and overall data for the developed ANN models.

| $R^2 = 1 - \left(\frac{\sum_i^N (A_i - Y_i)^2}{\sum_i^N (Y_i)^2}\right)$ | $R^2$ (For Training Data) | $R^2$ (For Confirmation Tests) | $R^2$ (For All Data) |
|---|---|---|---|
| Downward axial force model | 0.999981092 | 0.998622385 | 0.999846253 |
| Failure force model | 0.999722461 | 0.998734858 | 0.999634318 |

$A_i$, $Y_i$, $N$ and $i$ are the experimental value, predicted value, total number of experimental data and trial number, respectively. The formula is from [25].

**Table 8.** The amount of experimental data, predicted data and its error for the confirmation experiments.

| Sample No. | Measured Downward Axial Force (N) | Predicted Downward Axial Force (N) | Error of Model for Downward Axial Force (N) | Measured Failure Force (N) | Predicted Failure Force (N) | Error of Model for Failure Force (N) |
|---|---|---|---|---|---|---|
| 15 | 3310 | 3479 | 5% | 2709 | 2810 | 3.7% |
| 20 | 4720 | 4553 | 3.5% | 4746 | 4876 | 2.7% |
| 27 | 2930 | 3050 | 4.1% | 2123 | 2025 | 4.6% |
| 34 | 3320 | 3373 | 1.6% | 3982 | 4156 | 4.4% |

(Error of model = $\frac{\lvert A_i - Y_i\rvert}{A_i}\times 100$).

### 3.2.1. Effect of Tool Geometry Parameters on the Welding Force and Tensile Shear Force

In this section, the effect of the tool geometry parameters on the downward axial force during RAFSW process and the fracture force at tensile shear test was studied, according to the developed ANN models in the previous section. Generally, a compromise between the fracture force and the downward axial force yields the best condition to make the welds by RAFSW technique. The reason is that the mentioned condition makes that possible to use low capacity, cost-effective CNC machines with minimized clamping and fixturing that causes to have a low-cost RAFSW process [5]. Therefore, having a joint with high fracture force while keeping the downward axial force minimized would be desirable. Among the designed and tested tools, tool No. 21 met this condition. Therefore, the following studies were done around this condition.

Figure 7 depicts the correlation of the shoulder diameter and the shoulder groove depth with the downward axial force and the fracture force. The downward axial force significantly increased

by the increase of the shoulder diameter as shown in Figure 7a. It was due to the increase of shear, deformation and material friction during the welding process [13]. Therefore, higher axial forces are generated during the welding process. Moreover, the increase of the shoulder groove depth caused the slight increase of the downward axial force. This could be related to the increase of the amount of the material involved in the mixing process between the grooves of the tool and the workpiece material. This leads to increase of the friction between the tool and material, slightly [13]. According to Figure 7b, the impact of the shoulder groove depth on the fracture force was negligible, while the increase of the tool shoulder diameter caused the decrease of the fracture force. It could be associated to the fact that at higher shoulder diameters, the downward axial force was higher, which caused a higher heat input. Thus, the heat-affected zone (HAZ) around the nugget zone will be larger [18]. As the HAZ is the weakest place in the weld area in terms of strength, the larger HAZ causes a lower strength of the joint. Therefore, the fracture force decreases.

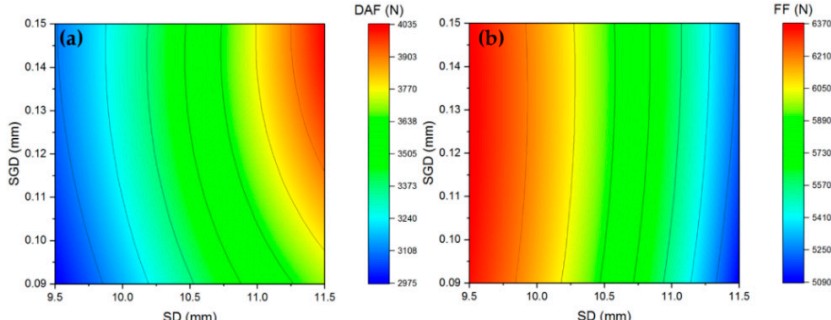

**Figure 7.** Contour plots of: (**a**) The downward axial force during the RAFSW process, and (**b**) failure force at tensile shear test versus shoulder diameter (*SD*), and shoulder groove depth (*SGD*), while other geometry and process parameters are the same as parameters of sample No. 21. The contour plots were extracted from developed ANN models in this paper.

As shown in Figure 8a, the increase of the pin length led to the increase of the downward axial force. It is attributed to the higher friction between the tool pin and the workpiece material during the mixing and stirring process due to larger interfacial area [18,20]. When the pin angle increased from 19° to 21°, the downward axial force increased at lower pin lengths. However, the axial force decreased by an increase of the pin angle at higher pin lengths. This behavior could be assigned to the interwoven relationship between the effect of pin angle and pin length on the material flow during the welding. Thus, it had a complex effect on the stirring and mixing mechanisms during the welding process. Figure 8b illustrates the correlation of the pin length, the pin angle and the fracture force. It indicates that the fracture force gradually increased when the pin angle increased from 19° to 21°. It could be due to the enhancement of the efficiency of stirring mechanism [18]. Moreover, the increase of the pin length led to the decrease of the fracture force. It can be due to the formation of more deteriorating hooking and cold lap defects. In fact, hooking and cold lap defects are the main cause of weakness of the FSW lap joints. The geometry and severity of these defects are affected by tool geometry and process parameters, considerably [18,20].

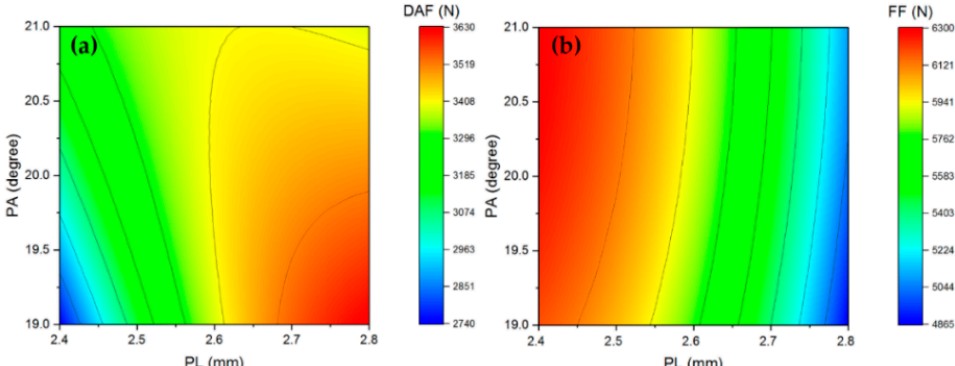

**Figure 8.** Contour plots of: (**a**) The downward axial force during the RAFSW process; (**b**) Failure force at a tensile shear test versus pin length (*PL*), and pin angle (*PA*) while other geometry and process parameters are the same as parameters of sample No. 21 except for the plunge depth, which is the same as the pin length. The contour plots were extracted from developed ANN models in this paper.

Figure 9 illustrates the impact of the pin base diameter and the pin lead on the downward axial force and the fracture force. It can be seen that the larger the pin base and the pin lead, the higher the downward axial force, as shown in Figure 9a. The reason is that the friction between the tool and workpiece increases by an increase of these parameters that eventually causes it to increase the axial force [18]. According to Figure 9b, increase of the pin base diameter and the pin lead resulted in increase of the fracture force. It can be attributed to the fact that at constant pin length and plunge depth, the increase of the pin base diameter and the pin lead can cause the formation of less defects in the joint. As a result, the fracture force boosts by these changes [18].

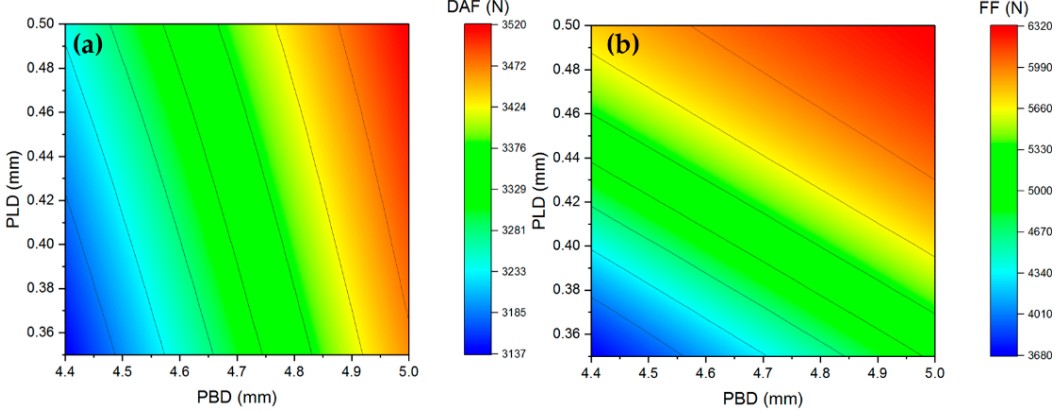

**Figure 9.** Contour plots of: (**a**) The downward axial force during the RAFSW process; (**b**) Failure force at a tensile shear test versus pin base diameter (*PBD*), and pin lead (*PLD*), while other geometry and process parameters are the same as parameters of sample No. 21. The contour plots were extracted from developed ANN models in this paper.

### 3.2.2. Effect of Process Parameters on the Welding Force and Tensile Shear Force

The correlation of the welding tool traverse speed and the tool rotational speed with the downward axial force is depicted in Figure 10a. It can be observed that the increase of the traverse speed led to an increase of the downward axial force. It is due to the fact that the more the traverse speed, the material in front of the tool would be colder [1]. Indeed, the higher the tool traverse speed, the less heat input would be generated in the weld area [1]. Thus, the needed axial force to pass the tool through the cold material of the workpiece would be high. In addition, Figure 10a indicates that the higher the rotational speed, the lower the downward axial force. This is due to the increase of the heat input at higher rotational speed that causes softer and warmer material ahead of the tool [1]. Figure 10b

illustrates that the increase of the tool rotational speed caused lower failure force. It could be due to the excessive heat input at high rotational speeds. The excessive heat input makes the HAZ area wider, which has an adverse effect on the strength of the joint [1]. Moreover, this figure shows that the increase of the tool traverse speed reduced the failure force by affecting the mixing and stirring processes, adversely. This results in the formation of defects and weak joints [12,21]. Compared to butt joints, lap joints strength highly depends on hook defects, which does not exist in butt joints. In butt joints, the fracture happens in the low hardness region, which is usually located in the HAZ area [30], while besides the weakness in the hardness of HAZ area in lap joints, other factors such as the hook defect, cold lap defect and tow crack-like sites play an important role in fracture of the lap joints. As a result, differences in the strength of the butt joint and lap joints can be observed in response to the variation of tool design and process parameters such as traverse speed [30].

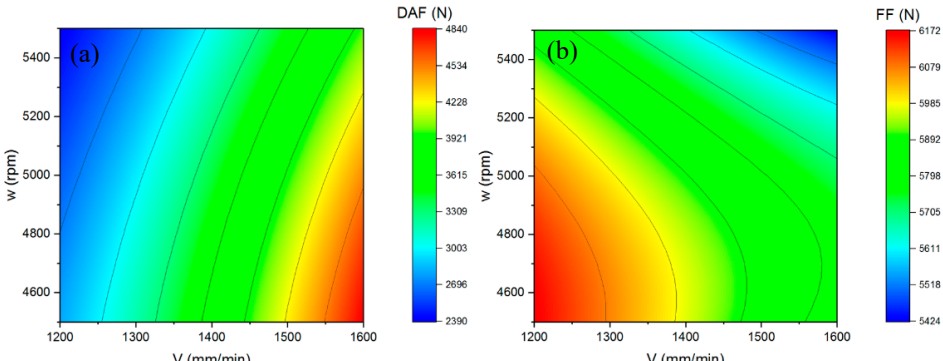

**Figure 10.** Contour plots of: (**a**) The downward axial force during the RAFSW process; (**b**) Failure force at a tensile shear test versus welding traverse speed (*V*) and welding rotational speed (*w*) while other geometry and process parameters are the same as parameters of sample No. 21. The contour plots were extracted from developed ANN models in this paper.

According to Figure 11a, the downward axial force increased with the plunge depth at a given tool traverse speed (when the pin length was the same as the plunge depth). Additionally, as shown in Figure 12a, a higher plunge depth increased the downward axial force at a given rotational speed. These are due to the higher frictional contact between the tool and the material at higher plunge depths [18,20]. Figures 11b and 12b also show that higher plunge depths lowered the fracture force, which can be attributed to the formation of the more damaging hooking defect and a large extent of upper sheet thinning in the weld area when the tool plunge depth is too high [9,12,22].

Generally, in both experimental results and ANN models, configuration 1 (when the advancing side is on the upper sheet) was preferable in terms of the failure force. This is in conjunction with reported research [10,11]. This is mainly due to the fact that the cold lap defect is not as deteriorating as the hooking defect [9]. When the upper sheet is on the advancing side, the cold lap defect is present in this side. Additionally, in all experiments, the fracture occurred in the welded region of the joint. This means that the reason of variation of joint strength is really related to the tool design and process parameters used for each weld.

As mentioned, a compromise between the fracture force and the downward axial force yields the best condition to make the welds by the RAFSW technique. Therefore, having high fracture force while keeping the downward axial force minimized would be desirable. Based on Figures 7–12, an efficient working window of the tool geometry and process parameters to make lap joints by the RAFSW technique is presented in Table 9.

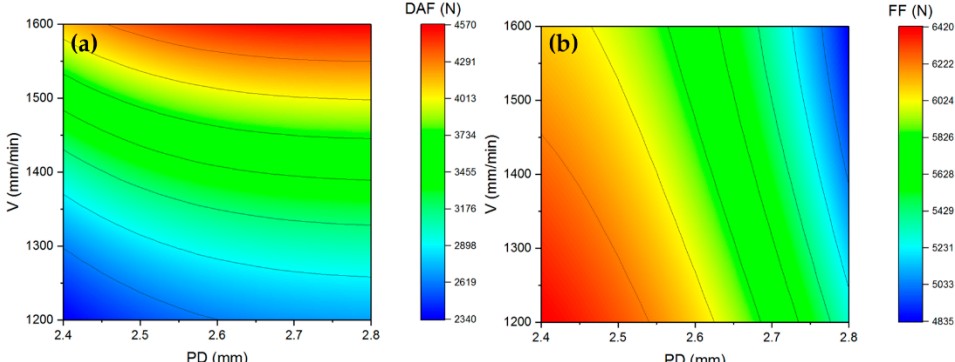

**Figure 11.** Contour plots of: (**a**) The downward axial force during the RAFSW process; (**b**) Failure force at tensile shear test versus tool plunge depth (*PD*) and welding traverse speed (*V*), while other geometry and process parameters are the same as parameters of sample No. 21 except for the plunge depth, which is the same as pin length. The contour plots were extracted from developed ANN models in this paper.

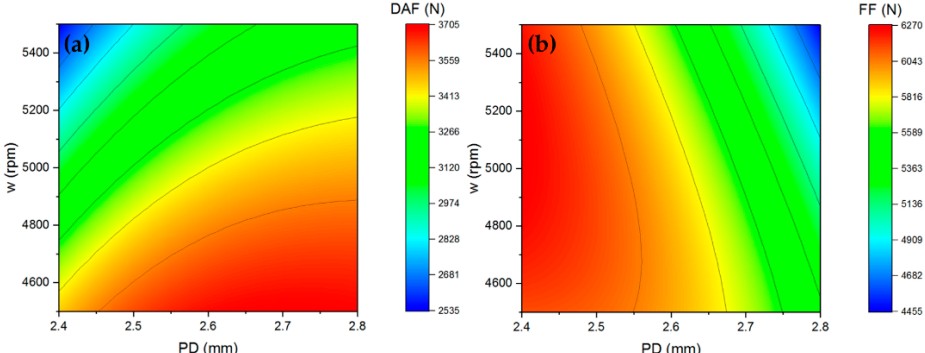

**Figure 12.** Contour plots of: (**a**) The downward axial force during the RAFSW process; (**b**) Failure force at a tensile shear test versus tool plunge depth (*PD*) and welding rotational speed (*w*) while other geometry and process parameters are the same as parameters of sample No. 21 except for the plunge depth, which is the same as pin length. The contour plots were extracted from developed ANN models in this paper.

**Table 9.** An efficient working window of the geometry and process parameters to make lap joints by RAFSW.

| PL (mm) | SD (mm) | SGD (mm) | PBD (mm) | PA (°) | PLD (mm) | V (mm/min) | W (rpm) | PD (mm) | C |
|---|---|---|---|---|---|---|---|---|---|
| 2.5–2.65 | 9.5–10.5 | 0.09–0.12 | 4.7–4.9 | 19–20 | 0.44–0.46 | 1200–1450 | 4500–5200 | 2.5–2.65 | 1 |

## 4. Conclusions

In the present study, RAFSW technique was applied to make AA6061-T6 lap joints. The main goal was to evaluate the effect of the tool design and the process parameters on the downward axial force generated during the welding process and the fracture force of the joints. The Taguchi method was used to minimize the number of experiments; and ANN modeling was implemented to anticipate the behavior of the joints. The main results of this study are presented as follows (all the statements are related to the studied range of the parameters in this paper):

- The effect of tool design and process parameters on the quality of the lap joints and the generated forces during the RAFSW process were predicted with high accuracy using ANN modeling.
- Larger shoulder diameters and pin lengths caused an increase of the downward axial force and a decrease of the fracture force of the joints.

- The downward axial force and the fracture force of the joints increased with an increase of pin base diameter and pin lead. However, the variation of shoulder groove depth and pin angle had minor effects on the axial force and the failure force, in the studied range.
- The increase of the tool rotational speed caused a reduction of the downward axial force and the fracture force. The downward axial force increased and the fracture force decreased with an elevation of tool traverse speed and rotational speed.
- The efficient range for tool design and process parameters to make quality lap joints at good traverse speeds is provided in Table 9.
- Making quality lap joints at high traverse speed while keeping the downward axial force as low as possible is the most promising condition for industrial users. According to this study, one could accomplish a sound, quality AA6061-T6 lap joint made by the RAFSW technique at traverse speeds as high as 1400 mm/min and downward axial forces as low as 3.2 kN.

The results of this study can be used as a roadmap to make quality AA6061-T6 lap joints by the RAFSW technique for industrial applications.

**Author Contributions:** M.M. and M.G. conceived and designed the experimental processes and made the welds; M.M. performed the characterization tests, M.M. conducted the ANN modeling and the analysis of the results; M.M. wrote the manuscript; M.G. supervised the experiments, modeling, and analysis of the results, and revised the manuscript.

**Funding:** This research has been supported by funds of $PI^2$ Team.

**Acknowledgments:** The authors would like to thanks the $PI^2$/REGAL team members who made this research possible.

**Conflicts of Interest:** The authors declare no conflict of interest.

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
