# Peer review of "Effect of Tool Design and Process Parameters on Lap Joints Made by Right Angle Friction Stir Welding (RAFSW)"

_jmmp, doi:10.3390/jmmp3030066_

Round 1
Reviewer 1 Report
This article presented an interesting research result about evaluating the effect of the tool design and the process parameters on the downward axial force generated during the welding process and the fracture force of the joints. Especially, artificial neural network modeling was implemented to anticipate the plunging force and the strength of the joints. The results showed that the artificial neural network developed in this study can accurately predict the plunging force and the strength of the joint. This method can be developed into a valuable tool for engineers working in similar tasks. Some minor changes were provided to the authors for consideration.
(1) The present title is too long and not very attractive although it clearly described what has been done in this study. Please consider something like “Welding forging force and joint strength prediction of friction stir welded lap joints using artificial neural network modeling” which may attract a broader range of readers.
(2) A previous publication “Influence of Tool Dimension and Welding Parameters on Microstructure and Mechanical Properties of Friction-Stir-Welded 6061-T651 Aluminum Alloy” in Metallurgical and Materials Transactions A 2008,39, 2378-2388.” showed that higher welding speed tended to lead higher joint strength for but joints for the metallurgical point of view. The authors need to compare the results of this article with the present study and put some lights on the difference in joint strength control for lap joints and but joints. Such as the consideration of hook effect….
(3) Please illustrate or describe the definition of “pin angle, pin base diameter, and pin lead”.
(4) Conclusion 1 can be removed as it is obvious without conducting the study and it did not add new value to the field.
(5) Please consider adding a conclusion about the effectivenesses of the artificial neural network on predicting welding forcing force and joint strength.”
Reviewer 2 Report
Title: Study the Effect of Tool Design and Process Parameters on Welding Axial Force and Strength of AA6061-T6 Lap Joints made by Friction Stir Welding at Right Angle (RAFSW)
Serial No.: jmmp-553482
This paper studied the effects of tool geometry and processing parameter on welding axial force and joint strength of right angle friction stir welds. The Taguchi method has been adopted for experimental design and artificial neural network (ANN) has been used for data analysis and performance prediction. The optimal tool design and processing parameter window were proposed. Overall, the results of this article are quite interesting and valuable. In my opinion the paper is worth publishing with the following corrections or complements:
1. Line 133: Please clarify the reason that the tool plunge depth in the second pass need to be deeper by 0.05 to 0.08 mm.
2. Table 2:
a. Please check the unit of DAF and FF. The unit here is kN but become N in the following contents and figures.
b. Some parameters of sample no. 27 are weird. Please check them carefully.
c. There are many tool parameters listed here. However, the definitions of some parameters cannot be understood easily. Please add one schematic plot for all the tool parameters to avoid confusion.
3. Line 171: Please clarify why the optimal learning rate and moment coefficient should be 0.5.
4. Line 172: Please clarify why the neuron number of the hidden layer should be 8.
5. Line Table 8: The measured DAF of sample no. 15 is wrong here. Please correct it.
6. Line 224: Please explain how to determine the sample no. 21 is the best choice.
7. Conclusions:
The final statement tries to mention how to make a quality lap joints. However, in the statement, only a transverse speed of 1400 mm/min can be seen. Any suggestion for a good tool design or good processing parameters cannot be found. Maybe the information from Table 9 can be mentioned here for readers.
Reviewer 3 Report
The present manuscript investigated the effect of tool design and parameters on the joint strength and axial force during the welding via Taguchi method and artificial neural network. The content is interesting, and I have a few suggestions for authors to make this paper more informative to readers.
Introduction:
(1) Authors should provide a more detailed description on the RAFSW technique rather than giving a lot of references. (Lines 32-33)
(2) FSW has also been applied for lap welding significantly for sure. (Line 33)
(3) There are many kinds of defects for lap welding. And hooks are not necessarily bad to joint strength. (Lines 45-47) Please check the paper as follows:
1. [Wang, T., Sidhar, H., Mishra, R.S., Hovanski, Y., Upadhyay, P. and Carlson, B., 2019. Effect of hook characteristics on the fracture behaviour of dissimilar friction stir welded aluminium alloy and mild steel sheets. Science and Technology of Welding and Joining, 24(2), pp.178-184.]
2. Linking process and structure in the friction stir scribe joining of dissimilar materials: A computational approach with experimental support
3. Investigation of Interfacial Layer for Friction Stir Scribe Welded Aluminum to Steel Joints
(4) Authors should provide some background of application of Al 6061-T6 alloy. (Lines 79-80)
Materials and Methods:
(1) Common tool dimensions such as pin length, shoulder diameter and specific tool dimensions such as pin base diameter, pin angle, pin lead should be labelled on Figure 1 (a).
(2) Label of 24 mm overlapping width was partly missing. (Figure 2 (b))
(3) Did the authors use some small pieces with the same thickness of welding sheet to balance the clamping load during lap shear testing? If so, please add the description in the manuscript.
Results and Discussion:
(1) Authors claimed that some separation occurred during friction stir lap welding, and the gap can be avoided by applying a rounded-edge tool. Can authors explain the reasons why rounded-edge tool can prevent the gap from occurring? (Lines 139-144)
(2) In tables 2 and 3, the unit of failure force was wrongly typed in kN, which should be N.
(3) Authors only correlated the joint strength with tool dimensions and process parameters, which is not accurate. Fracture location of each joint should be provided. Let us assume one condition: All the joints broke at base materials when the welded interface is strong enough. Then the joint strength will not vary with the tool dimensions and process parameters.
